# Real-Time Detection of Incipient Inter-Turn Short Circuit and Sensor Faults in Permanent Magnet Synchronous Motor Drives Based on Generalized Likelihood Ratio Test and Structural Analysis

**DOI:** 10.3390/s22093407

**Published:** 2022-04-29

**Authors:** Saeed Hasan Ebrahimi, Martin Choux, Van Khang Huynh

**Affiliations:** Department of Engineering Sciences, University of Agder, 4879 Grimstad, Norway; martin.choux@uia.no (M.C.); huynh.khang@uia.no (V.K.H.)

**Keywords:** fault diagnosis, inter-turn short circuit, sensor fault, structural analysis, generalized likelihood ratio test, PM synchronous motor

## Abstract

This paper presents a robust model-based technique to detect multiple faults in permanent magnet synchronous motors (PMSMs), namely inter-turn short circuit (ITSC) and encoder faults. The proposed model is based on a structural analysis, which uses the dynamic mathematical model of a PMSM in an abc frame to evaluate the system’s structural model in matrix form. The just-determined and over-determined parts of the system are separated by a Dulmage–Mendelsohn decomposition tool. Subsequently, the analytical redundant relations obtained using the over-determined part of the system are used to form smaller redundant testable sub-models based on the number of defined fault terms. Furthermore, four structured residuals are designed based on the acquired redundant sub-models to detect measurement faults in the encoder and ITSC faults, which are applied in different levels of each phase winding. The effectiveness of the proposed detection method is validated by an in-house test setup of an inverter-fed PMSM, where ITSC and encoder faults are applied to the system in different time intervals using controllable relays. Finally, a statistical detector, namely a generalized likelihood ratio test algorithm, is implemented in the decision-making diagnostic system resulting in the ability to detect ITSC faults as small as one single short-circuited turn out of 102, i.e., when less than 1% of the PMSM phase winding is short-circuited.

## 1. Introduction

Permanent magnet synchronous motors (PMSMs) have gained popularity in industrial applications such as electric vehicles, robotic systems, and offshore industries due to their merits of efficiency, power density, and controllability [1,2,3]. PMSMs working in such applications are constantly exposed to electrical, thermal, and mechanical stresses, resulting in different faults such as electrical, mechanical, and magnetic faults [4]. Among these various faults, the stator winding inter-turn short circuit (ITSC) fault is considered as one of the most common faults [5] due to the excessive heat produced by a high circulating current in a few shorted turns of the stator winding [6]. Subsequently, this excessive heat causes further insulation degradation and might lead to a complete machine failure [7] if it is not detected and treated in time. Therefore, developing methods for monitoring and detecting the ITSC fault in its early stages can substantially lower maintenance costs, downtime of the system, and productivity loss.

ITSC faults can be detected by signal-based, data-driven, and model-based techniques [8]. The first approach aims to detect fault characteristic frequencies in measured motor signals, namely, current, voltage, or vibration signals [9,10,11], being processed by time–frequency signal analysis tools such as Fourier transform [12], matched filters [13], Hilbert–Haung transform [14], wavelet transforms [15], and Cohen distributions [16]. These signal-based methods face challenges of real-time implementations due to the computational burden, and missing fault characteristic signals does not guarantee that the machine is healthy [8]. The data-driven approach such as an artificial neural network (ANN) [17] and Fuzzy systems [18] requires a lot of historical data to train models and classify localized faults. Historical data is restricted in industry and producing a lot of historical data in healthy and faulty conditions is costly and time-demanding [19]. Alternatively, model-based techniques have been proposed to detect ITSC faults [20,21,22]. Among them, the finite element method (FEM)-based models have been widely used due to the accuracy and convenience of taking into account physical phenomena, e.g., saturation. FEM models, known as time-demanding and computationally heavy ones, require deep knowledge of the system, e.g., detailed dimensions and material characteristics. Other model-based methods that use mathematical equations to model a motor’s behavior have been reported to have challenges regarding validity when experiencing abnormal conditions such as internal faults [8]. To address the mentioned challenges, structural analysis is proposed as an alternative solution for detecting ITSC faults in electrical motors. The structural analysis algorithm has been well studied and developed in the literature [23,24,25] and applied to different structures. The structural analysis approach has been able to successfully detect faults in automotive engines [26,27,28], hybrid vehicle [29], and battery systems [30]. In [31,32]. The algorithm has successfully been applied on PMSM electric drive systems to detect sensor faults such as voltage, current, encoder, and torque sensors. In our previous study [33], it was proposed that the algorithm can be used on an electric drive system to also detect common physical faults in PMSMs such as ITSC and demagnetization, and residual responses were obtained by simulation. However, in previous studies, this algorithm has not been implemented in real-time diagnosis of an industrial PMSM for detection of ITSC faults. Implementing a structural analysis technique on a PMSM and drive, this paper aims to achieve the following contributions:Detection of both internal motor faults and external measurement faults, namely ITSC and encoder faults;Detection of the lowest level of ITSC fault, with one shorted turn in stator phase winding;Early detection of an ITSC fault, i.e., considering a lower fault current in the degradation path as compared to shorted turns;Modeling of the noise in drive system measurement signals with unknown amplitude and variance.

This paper presents a systematic fault diagnosis methodology based on structural analysis for detecting multiple faults in PMSM drives, namely ITSC faults and encoder fault. To achieve this, a healthy dynamic mathematical model of PMSM is defined in the abc reference frame based on the dynamic constraints, measurements, and derivatives. To model an ITSC fault in any phase, specific fault terms are added to the three-phase flux and voltage equations. These fault terms include the deviations in the voltage, flux, and currents of the stator winding caused by an ITSC fault, since a part of winding is shorted; hence, three-phase voltage and flux signals are subjected to changes. In addition, fault terms are added to the dynamic model to take into account the encoder faults, resulting in errors of the angular speed and angle measurements. Subsequently, the analytical redundant part of the structural model is extracted and divided into minimally over-determined sub-systems from which three sequential residuals are obtained based on the error in the current signal of each phase. Furthermore, a resultant residual is formed in the αβ frame to achieve a better demonstration of different ITSC fault levels. Finally, a generalized likelihood ratio test is developed to detect the faults in the resultant and encoder residuals under unknown noise parameters assumptions, i.e., unknown amplitude and unknown variance.

## 2. Modeling Inter-Turn Short-Circuit Fault

The studied PMSM consists of distributed three-phase windings on the stator and PMs on the rotor. Each phase winding contains several coils in parallel, being formed by wrapping bundles of wires together. The wire insulation of the stator windings might be degraded over time under electrical, mechanical, and thermal stresses, which may eventually lead to electrical faults such as an inter-turn short circuit (ITSC), a phase to ground short circuit (PGSC), and a phase to phase short circuit (PPSC).The stator ITSC fault is considered the most common electrical fault [34] and usually occurs in a few shorted turns. The degraded path among the shorted turns is provided by a nonzero resistance of the faulty insulation, leading to a circulating fault current. This circulating fault current results in copper losses and excessive heat in the shorted turns since only a few turns are involved, and the current-limiting impedance is low. The insulation might further degrade and even propagate to nearby turns. This might cause other critical faults such as a PGSC fault, a PPSC fault, and even a complete failure. Therefore, monitoring and detecting the ITSC fault in early stages would reduce costs and downtime caused by the machine failure.

To model ITSC faults in a PMSM, it is necessary to know how the motor signals and parameters are affected by the different levels of the fault. The schematic of a PMSM stator winding under ITSC faults with different levels in each phase is shown in Figure 1. The level of fault in abc phases is denoted by $\mu_{a}$, $\mu_{b}$, and $\mu_{c}$, respectively, which are defined by the ratio of the number of shorted turns to the total number of turns per abc phase winding. In a healthy condition, each phase winding of a PMSM has a resistance of $R_{s}$ and an inductance of $L_{s}$. In the presence of ITSC faults in each phase, the phase winding is split into a faulty part with $\mu R_{s}$ and $\mu L_{s}$, and a healthy part with $\left(1-\mu \right)R_{s}$ and $\left(1-\mu \right)L_{s}$ resistance and inductance values. As a result, there is not only mutual inductance between healthy and faulty parts in each phase winding, but also between the faulty winding with other phase windings [35]. In addition, the degraded resistance of the insulation in each phase is denoted by $R_{af}$, $R_{bf}$, and $R_{cf}$, while the circulating fault currents are $i_{af}$, $i_{bf}$, and $i_{cf}$, respectively. To detect an incipient ITSC fault, the resistance of the degraded path should be higher than the resistance of the shorted turns [36]. This is due to the fact that an ITSC fault forms gradually over time and starts with a low current circulating through the degraded path.

## 3. Structural Analysis for PMSM with ITSC and Encoder Faults

Structural analysis aims to extract the analytic redundant relations (ARRs) of a system based on the mathematical equations that describe the system’s dynamic [23,37]. A structural analysis algorithm relies on redundancy in a system (a redundant part of the complex system) and yields residuals for fault detection and isolation (FDI) based on ARRs. Assuming that a model *M* has outputs *z* and inputs *u*, a residual is extracted by eliminating all the unknown variables, i.e., substituting an unknown variable with its equivalent obtained value through a redundant path. Therefore, it leads to a relation that contains only the known variables r(u,z)=0 which is known as an ARR if the observation *z* is consistent with the system model [23]. As a result, this residual’s response will maintain a zero value under a null hypothesis (nonfaulty case) $H_{0}$ and a nonzero value under an alternative hypothesis (faulty case) $H_{1}$ as follows:(1)$$\begin{array}{c} H_{0}:r\left(u,z\right)=0\\ H_{1}:r\left(u,z\right)\neq 0 \end{array}$$This methodology is especially effective for fault diagnosis of complex systems where a prior deep knowledge of the whole system is neither needed nor affordable in terms of computational burden and processing time. Instead, a small redundant part of the system is selected and processed to obtain smaller redundant subsystems that can be used in forming residuals for detecting each predefined fault. First, the structural model of a redundant system is formed and represented by an incidence matrix with variables as columns and equations as rows. The variables are categorized as unknown variables, known variables, and faults, while the equations are categorized as dynamic equations, measurements, and differential equations. Each row of the incidence matrix connects an equation to the corresponding variables if they are present in that specific equation. Next, the just-determined and over-determined parts of the system are separated by rearranging the rows and columns in a way to form a diagonal structure that is known as Dulmage–Mendelsohn (DM) decomposition. Using the analytic redundant part of this structure and based on the degree of the redundancy, several smaller sets of ARRs are identified. These smaller sets are called minimally over-constrained sets and have one degree of redundancy, holding exactly one more equation than the number of variables. Subsequently, a fault signature matrix is formed to demonstrate which fault can be detected or even discriminated. Finally, specific diagnostic tests (residuals) are designed to detect faults. Here, a structural analysis of a PMSM experiencing independent ITSC faults in each phase is presented, and diagnostic tests are proposed to detect and discriminate them. Figure 2 shows the modeling diagram of a faulty PMSM and the drive system where measurements are acquired by sensors and faults are located inside the motor.

### 3.1. PMSM Mathematical Model

The dynamic equations of a faulty PMSM in an abc frame with ITSC faults present in three phases are represented by equations $e_{1}-e_{9}$ as shown in (Equation 2), where $v_{a}$, $v_{b}$, and $v_{c}$ are the stator phase voltages; $i_{a}$, $i_{b}$, and $i_{c}$ are the stator phase currents; $\lambda_{a}$, $\lambda_{b}$, and $\lambda_{c}$ are the stator phase fluxes; $T_{e}$ is the electromagnetic torque; $T_{L}$ is the load torque; $\omega_{m}$ is the rotor’s angular speed; θ is the electric angular position; $R_{a}$, $R_{b}$, and $R_{c}$ are the stator phase resistances and $L_{a}$, $L_{b}$, and $L_{c}$ are the stator phase inductances; $\lambda_{m}$ is the flux produced by rotor PMs; *p* is the pole pairs; *J* is the rotor inertia, and *b* is the friction coefficient.

As discussed in Section 2, an ITSC fault splits the phase winding into a faulty part with resistance and inductance of $\mu R_{s}$ and $\mu L_{s}$ and a healthy part with resistance and inductance of $\left(1-\mu \right)R_{s}$ and $\left(1-\mu \right)L_{s}$. The changed resistance and inductance of the winding have direct correlation with voltage equations and flux equations. Under a healthy condition, the model of PMSM, especially $e_{1}$–$e_{6}$, have no fault terms. Therefore, any changes in the inductance will affect both voltage and flux equations ($e_{1}$–$e_{6}$) directly, and any changes in the resistance will affect only voltage equations ($e_{1}$–$e_{3}$) directly. Here, $f_{v_{a}}$ and $f_{\lambda_{a}}$ are added to the corresponding equations of the healthy PMSM to account for the ITSC fault in phase *a*. Similarly, $f_{v_{b}}$, $f_{v_{c}}$, $f_{\lambda_{b}}$, and $f_{\lambda_{c}}$ terms are added to account for ITSC faults in phases *b* and *c*, respectively. These fault terms are shown in red in (Equation 2).
(2)$$\begin{array}{cc} e1: & v_{a}=R_{a}i_{a}+\frac{d\lambda_{a}}{dt}+f_{v_{a}}\\ e2: & v_{b}=R_{b}i_{b}+\frac{d\lambda_{b}}{dt}+f_{v_{b}}\\ e3: & v_{c}=R_{c}i_{c}+\frac{d\lambda_{c}}{dt}+f_{v_{c}}\\ e4: & \lambda_{a}=L_{a}i_{a}+\lambda_{m}\cos\theta +f_{\lambda_{a}}\\ e5: & \lambda_{b}=L_{b}i_{b}+\lambda_{m}\cos\left(\theta -2\pi /3\right)+f_{\lambda_{b}}\\ e6: & \lambda_{c}=L_{c}i_{c}+\lambda_{m}\cos\left(\theta +2\pi /3\right)+f_{\lambda_{c}}\\ e7: & T_{e}=-p\lambda_{m}[i_{a}\sin\theta +i_{b}\sin\left(\theta -2\pi /3\right)\\ \; & +i_{c}\sin\left(\theta +2\pi /3\right)]\\ e8: & \frac{d\omega_{m}}{dt}=\frac{1}{J}\left(T_{e}-b\omega_{m}-T_{L}\right)\\ e9: & \frac{d\theta }{dt}=p\omega_{m} \end{array}$$

The known variables consist of the motor signals, which are measured for both control purposes and fault diagnosis. Thus, in addition to the three-phase currents and angular position, i.e., $y_{i_{a}}$, $y_{i_{b}}$, $y_{i_{c}}$, and $y_{\theta }$ that are necessary for the control system. Three-phase voltages, i.e., $y_{v_{a}}$, $y_{v_{b}}$, and $y_{v_{c}}$, are also measured to complete the diagnostic system. Equation (Equation 3) shows these known variables, where $f_{\theta }$ and $f_{\omega }$ fault terms are also added to account for speed and angle measurement error.
(3)$$\begin{array}{cc} m1:y_{v_{a}} & =v_{a}\;m4:y_{i_{a}}=i_{a}\;m7:y_{\theta }=\theta +f_{\theta }\\ m2:y_{v_{b}} & =v_{b}\;m5:y_{i_{b}}=i_{b}\;m8:y_{\omega_{m}}=\omega_{m}+f_{\omega_{m}}\\ m3:y_{v_{c}} & =v_{c}\;m6:y_{i_{c}}=i_{c} \end{array}$$

In addition, since the dynamic model of PMSM includes five differential constraints in the abc frame, these are needed to be defined as unknown variables. Equation (Equation 4) shows the differential constraints for the structural model.
(4)$$\begin{array}{cc} d1: & \frac{d\lambda_{a}}{dt}=\frac{d}{dt}\left(\lambda_{a}\right)\;d4:\frac{d\omega_{m}}{dt}=\frac{d}{dt}\left(\omega_{m}\right)\\ d2: & \frac{d\lambda_{b}}{dt}=\frac{d}{dt}\left(\lambda_{b}\right)\;d5:\frac{d\theta }{dt}=\frac{d}{dt}\left(\theta \right)\\ d3: & \frac{d\lambda_{c}}{dt}=\frac{d}{dt}\left(\lambda_{c}\right) \end{array}$$

### 3.2. Structural Representation of the PMSM Model

The structural model of the PMSM with ITSC and encoder faults is obtained based on the redundant dynamic model in (Equation 2)–(Equation 4), as shown in Figure 3. The incidence matrix contains 22 rows, representing the nine defined equations in (Equation 2), the eight measured known variables in (Equation 3), and the five differential constraints of unknown variables as shown in (Equation 4). The columns of the matrix are subdivided into three groups of unknown variables, known variables, and faults. The known variables are obtained directly from the measurements, while the unknown variables can be calculated based on the known variables. The faults considered in the structural model are variations in phase voltage and flux to represent ITSC faults in each phase.

### 3.3. Analytical Redundancy of the Model

To detect specific faults in a redundant system, faults must first be introduced to the model, and then a proper diagnostic test containing the considered fault is selected. A diagnostic test is a set of equations (or consistency relations) extracted from the system model, in which at least one equation is violated in case of the presence of a considered fault. A system model is called a redundant model if the system model consists of more equations than unknown variables. Assuming that model M=(C,Z) contains constraints (equations) *C* and variables *Z*, let unknown variables *X* be the subset of all variables *Z* in model *M* (X⊆Z). The degree of redundancy of the model *M* is defined as:(5)$$\begin{array}{c} \varphi (M)=|C|-|X| \end{array}$$
where |C| denotes the number of equations, and |X| is the number of unknown variables contained in the model *M*. According to bipartite graph theory, any finite dimensional graph such as M=(C,Z) can be decomposed into three sub-graphs as follows [37]:$M^{-}$: structurally under-determined part of the model *M*, where fewer equations than unknown variables lie, and the degree of redundancy is negative φ(M)<0.$M^{0}$: structurally just-determined part of the model *M*, where equal equations and unknown variables lie, and the degree of redundancy is zero φ(M)=0.$M^{+}$: structurally over-determined part of the model *M*, where more equations than unknown variables lie, and the degree of redundancy is positive φ(M)>0.

For diagnostic purposes, the over-constrained sub-graph is the interesting part because it contains the important redundancy that is necessary for detecting a fault. According to [38], a fault is structurally detectable if the equation that contains the fault variable lies in the over-determined part of the whole model ($e_{f}\in M^{+}$). To obtain these sub-graphs, a canonical decomposition of the main structure graph (M) is required. An example of this canonical decomposition is shown in Figure 4, where three canonical sub-models of the model *M* are obtained as {$M^{-}$, $M^{0}$, $M^{+}$}.

This canonical decomposition is achieved after the rows and the columns of the main structural graph (structural model incidence matrix) are rearranged so that the matched variables and constraints appear on the diagonal. Therefore, having a decomposition tool that analyzes the redundancy of the structural model and forms this diagonal structure is very beneficial. Dulmage–Mendelsohn (DM) is a key decomposition tool that is applied on a structural model directly and obtains a unique diagonal structure by a clever reordering of equations and variables [39]. Figure 5 shows the DM decomposition for the PMSM structural model, where the analytic redundant part is expressed in the bottom-right part containing all the faults. Since this part includes redundancy and can be monitored, diagnostic tests can be designed with the set of ARRs in $M^{+}$. As a result, if a fault is defined in the model and is supposed to be detected by the diagnosis system, a residual that is sensitive to the presence of that fault must exist.

## 4. Diagnostic Test Design

This section presents the procedure of designing diagnostic tests for ITSC and encoder faults. First, the over-determined part of the structural model is separated into smaller redundant subsystems where faults are observable, and then the sequence of obtaining residuals for the detection of each fault is explained.

### 4.1. Minimal Testable Sub-Models

According to the definition given by [38], an equation set *M* is a TES set if:F(M)≠⌀.*M* is a proper structurally over-determined set.For any $M^{\prime }⊋M$ where $M^{\prime }$ is a proper structurally over-determined set, it holds that $F\left(M^{\prime }\right)⊋F\left(M\right)$
where F(M) is the set of faults that influence any of the equations in *M*. A TES *M* is a minimal test equation support (MTES) if there exists no subset of *M* that is a TES holding the degree of redundancy of one. Following the algorithm in [38], the structural model is subdivided into efficient redundant MTES sets. Each MTES set contains a group of ARRs that together hold the degree of redundancy of one, meaning that there is only one equation more than the number of variables involved. In addition, they are obtained in a way that the effect of faults is considered. This reduces computational complexity significantly without reducing the possible diagnosis performance as compared to structurally over-determined (MSO) sets. Figure 6 shows all the MTES sets found for the considered structural model here, where each row of the matrix connects the corresponding MTES to the equations involved.

Figure 7 shows the signature matrix of MTES sets, indicating which fault terms are included in each MTES. $MTES_{1}$ includes $f_{\theta }$ and $f_{\omega }$ fault terms that can be used for detecting a rotor’s speed and angle measurement error. $MTES_{2}$ and $MTES_{3}$ contain $f_{v_{c}}$ and $f_{\lambda_{c}}$ fault terms for detecting an ITSC fault in phase *c*. $MTES_{4}-MTES_{6}$ contain $f_{v_{b}}$ and $f_{\lambda_{b}}$ fault terms for detecting an ITSC fault in phase *b*. Similarly, $MTES_{7}-MTES_{10}$ can be used for detecting an ITSC fault in phase *a* since it has $f_{v_{a}}$ and $f_{\lambda_{a}}$ fault terms. If a MTES set containing the information of changes in voltage and flux of a phase winding is found, it can be used to form a residual that is sensitive to the presence of an ITSC fault in that phase.

### 4.2. Diagnosability Index

An important criterion for selecting MTES sets is to satisfy diagnosability requirements. This includes detectability of any single fault as well as isolability between any two faults. Here, an index for the proper selection of MTES sets that are suitable to be used in sequential residual generators is introduced. Zhang [40] proposed a diagnosability index that is aimed at achieving the maximum degree of diagnosability for each residual by comparing the distance between the fault signature matrices of MTES sets:(6)$$\begin{array}{c} m_{D}=\frac{1}{\left(\begin{array}{c} n+1\\ 2 \end{array}\right)}\sum_{i=0}^{n-1}\sum_{j=i+1}^{n}D\left(V_{f_{i}},V_{f_{j}}\right) \end{array}$$
where $D(V_{f_{0}},V_{f_{j}})$ stands for the distance between the fault signature of $f_{j}$ and the healthy case and measures the detectability of fault $f_{j}$. $D(V_{f_{i}},V_{f_{j}})$ is the distance between two fault signatures and is defined as the Hamming distance [41] between the two fault signature strings:(7)$$\begin{array}{c} D\left(V_{f_{i}},V_{f_{j}}\right)=\sum_{n=1}^{S}\left|V_{f_{i}}-V_{f_{j}}\right| \end{array}$$

### 4.3. Sequential Residuals for Detecting ITSC and Encoder Faults

This section presents the sequence of deriving four residuals ($R_{1}-R_{4}$) based on the obtained MTES sets. These residuals aim to detect ITSC faults in any of the phase windings as well as encoder measurement faults. To form residual $R_{1}$ that is sensitive to an ITSC fault in phase *a* winding, an MTES set should be chosen that contains $f_{v_{a}}$ and $f_{\lambda_{a}}$ fault terms. As can be seen in the fault signature matrix in Figure 7, $MTES_{7}-MTES_{10}$ can be used for forming such a residual because these four MTES sets all contain $f_{v_{a}}$ and $f_{\lambda_{a}}$ fault terms. Among them, an MTES set is preferred that contains a lower number of fault terms because it will be more isolated and less influenced by other faults. $MTES_{7}$ and $MTES_{8}$ contain three fault terms, while $MTES_{9}$ and $MTES_{10}$ contain four fault terms. Therefore, either $MTES_{7}$ or $MTES_{8}$ should be chosen, and $MTES_{8}$ is preferred due to a lower number of involved equations ($MTES_{8}$ contains six equations, while $MTES_{7}$ contains eight equations) which leads to less complexity, as seen in Figure 6. $MTES_{4}-MTES_{6}$ can be used for forming residual $R_{2}$ because they contain $f_{v_{b}}$ and $f_{\lambda_{b}}$ fault terms. Among them, $MTES_{5}$ is preferred because it contains a lower number of fault terms compared to $MTES_{6}$ and a lower number of equations compared to $MTES_{5}$. Similarly, $MTES_{3}$ is chosen to form residual $R_{3}$ that is sensitive to an ITSC fault in phase-c winning and contains a lower number of equations compared to $MTES_{2}$, given the fact that both contain $f_{v_{c}}$ and $f_{\lambda_{c}}$ fault terms. To form residual $R_{3}$ that is sensitive to an encoder fault (angular velocity and position measurements), an MTES set is preferred that contains both $f_{\theta_{e}}$ and $f_{\omega_{m}}$, and the only MTES set that contains such fault terms is $MTES_{1}$. The combination of these four MTES sets, i.e., $MTES_{1}$, $MTES_{3}$, $MTES_{5}$, and $MTES_{8}$ yield a high diagnosability index as $m_{D}=1.88$, and this maximizes the chance of discrimination of each fault from others. The sequential residuals are obtained as follows:$R_{1}$: $MTES_{8}$ is used for deriving $R_{1}$ based on the error between calculated and measured current of phase *a* winding, i.e $m_{4}$ in (Equation 3):
(8)$$\begin{array}{c} m_{4}:R_{1}=i_{a}-y_{i_{a}} \end{array}$$And the sequence of obtaining these variables is as follows:
(9)$$\begin{array}{cc} SV_{1}:\lambda_{a} & =\lambda_{a_{state}}\\ m_{7}:\theta & =y_{\theta }\\ m_{1}:v_{a} & =y_{v_{a}}\\ e_{4}:i_{a} & =\frac{1}{L_{a}}\left(\lambda_{a}-\lambda_{m}\cos\theta \right)\\ e_{1}:d\lambda_{a} & =v_{a}-R_{a}i_{a} \end{array}$$
where $\lambda_{a_{state}}$ is a state variable and updated at each time-step as follows:
(10)$$\begin{array}{c} e_{17}:\lambda_{a_{state}}=\int d\lambda_{a}dt \end{array}$$$R_{2}$ and $R_{3}$ follow the same procedure mentioned for $R_{1}$ based on the error between calculated and measured currents of phase *b* and phase *c* using $MTES_{5}$ and $MTES_{3}$, respectively.$R_{4}$: $MTES_{1}$ is used for deriving $R_{4}$ based on the error between the calculated and measured shaft’s angular speed, i.e $m_{8}$ in (Equation 3):
(11)$$\begin{array}{c} m_{8}:R_{4}=\omega_{m}-y_{\omega_{m}} \end{array}$$
and the sequence of obtaining the unknown variable, $\omega_{m}$, is as follows:
(12)$$\begin{array}{cc} m_{7}:\theta & =y_{\theta }\\ d_{5}:\frac{d\theta }{dt} & =\frac{d}{dt}\left(\theta \right) \end{array}$$
(13)$$\begin{array}{cc} e_{9}:\omega_{m} & =p\frac{d\theta }{dt} \end{array}$$

## 5. Experiments and Results

The proposed diagnostic method is implemented and validated through an in-house experimental setup in this section. First, ITSC faults were applied to the phase windings of a four-pole PMSM, as shown in Figure 8. Each phase winding of the motor has two coils in series, each of which has 51 turns with three parallel branches. For phase *a*, one of the turns was short circuited, or about a 1% fault level. For the phases *b* and *c*, three and five turns were short circuited, resulting in almost 3% and 5% fault severity, respectively. The connection wires to these extra taps in the phase windings were taken out of the motor and connected to 100 mΩ resistors (similar to $R_{f}$ Figure 1) both to limit the short circuit current and to simulate the winding insulation degradation, as shown in Figure 9. Furthermore, controllable relays were placed between winding taps and fault resistors to activate or deactivate the fault. The faulty motor was mechanically coupled to a generator as a variable load and an incremental encoder to measure the rotor’s angle and velocity. The motor was driven by a Watt&Well DEMT 3-ph voltage source inverter, which had embedded voltage and current sensors, being fed by a Keysight N8949A dc supply. In addition, a dSpace MicrolabBox control unit was used as a real-time interface device for implementing both control strategy and data acquisition from Matlab/Simulink with a sampling time of 50 μs. The parameters of the studied PMSM are listed in Table 1.

To test the residual responses and effectiveness of the diagnostic system, the motor was driven from stationary to nominal speed, i.e., 1500 rpm, and kept in a steady-state condition. During the operation of the motor, the encoder and ITSC faults were applied at different time intervals using controllable relays. At *t* = 1–3 s, the encoder measurement fault was applied with a 1 rad/s error. At *t* = 4.471–7.238 s, the ITSC fault in phase *a* was applied which had 1% fault severity (one shorted turn in phase *a* winding); at *t* = 9.613–12.76 s, the ITSC fault in phase *b* appeared with 3% fault severity (three shorted turn in phase *b* winding); at *t* = 15.6–18.41 s, the ITSC fault in phase *c* with 5% fault severity (five shorted turn in phase *c* winding) was applied on the motor.

The residual responses for the mentioned faults were obtained and are shown in Figure 10. Before the faults were applied, the motor was operating in a healthy mode (*t* = 0–1 s), and all the residuals remained averagely zero (neglecting the noise). This is because there was no error between the measured signals and the calculated ones used in each residual. First, when the encoder fault appeared, $R_{4}$ obtained a nonzero dc value, and it went back to average zero as soon as the fault disappeared. When the ITSC fault in phase *a* was applied, $R_{1}$ was directly affected and obtains/ed a higher oscillating value. Due to mutual induction of the fault current, this fault was also observable in $R_{2}$ and $R_{3}$. In addition, the controller response had a role in the increase of other phase currents. Since a part of the winding was gone, more $I_{q}$ was required to keep the motor speed constant at 1500 rpm. The same logic can be used for ITSC faults in phases *b* and *c* as the residuals obtain higher oscillating values. The behavior and response of the residuals during each ITSC fault, can be used as the ground for detection of faults in the PMSM. This is implemented using signal processing–detection theory and explained in the following section.

## 6. Diagnostic Decision

Using the residual responses, a diagnostic decision making system was designed to detect the ITSC faults based on statistical signal processing–detection theory. While $R_{4}$ can be directly used to detect encoder faults, a combination of $R_{1}$–$R_{3}$ is required to effectively detect ITSC faults. The $R_{1}$–$R_{3}$ residuals obtained in the previous section, are designed based on abc frame voltage equations $e_{1}$–$e_{3}$ in (Equation 2), and an ITSC fault in any phase creates unbalance in the residual output. Before designing the statistical detector and to form a better index that obtains a nonzero dc value in case of an ITSC fault, the residuals in the abc frame are taken into an αβ frame using the power invariant Clarke transformation as follows:(14)$$\left[\begin{array}{c} R_{\alpha }\\ R_{\beta } \end{array}\right]=\sqrt{\frac{2}{3}}\left[\begin{array}{ccc} 1 & -\frac{1}{2} & -\frac{1}{2}\\ 0 & \frac{\sqrt{3}}{2} & -\frac{\sqrt{3}}{2} \end{array}\right]\left[\begin{array}{c} R_{1}\\ R_{2}\\ R_{3} \end{array}\right]$$

The absolute value of the resultant is calculated:(15)$$\begin{array}{c} R_{r}=\left|R_{\alpha }+jR_{\beta }\right| \end{array}$$

Figure 11 shows the absolute value of the resultant residual in an αβ frame where ITSC faults in all phases are more obvious compared to abc residuals $R_{1}$–$R_{3}$.

In implementing a structural analysis, the goal was to form residuals that have a zero value in a healthy scenario and a nonzero value in a faulty scenario. However, derivatives, integrals, and even uncertainties in the dynamic model affect the calculation of unknown variables and cause the variable output signal to be a little bit distorted. In addition, phenomena such as environmental noise and switching noise affect the signals. These lead to a residual output signal that fluctuates around zero instead of having a perfect signal that holds the absolute zero value in a healthy scenario. Even in a faulty scenario, the residual signal fluctuates around a nonzero value as seen in Figure 11. Therefore, extra signal processing is required to deal with model uncertainties and environmental noise and to be able to distinguish and isolate the indicator signal from noise. Here, a generalized likelihood ratio test (GLRT) is proposed to deal with such model uncertainties and also to provide the ground for calculating and setting thresholds based on the probabilities of detection and false alarms in a formulated and scientific manner.

### 6.1. Generalized Likelihood Ratio Test

GLRT is a composite hypothesis testing approach that can be used for detecting a signal in realistic problems [42]. It is noted that GLRT does not require prior knowledge of the unknown parameters such as mean (μ) and variance ($\sigma^{2}$) values in a probability density function (PDF) of a signal. GLRT deals with unknown parameters by replacing them with their maximum likelihood estimates (MLEs). If data *x* have the PDF $p(x;\hat{\theta_{0}},H_{0})$ under a null hypothesis $H_{0}$ and $p(x;\hat{\theta_{1}},H_{1})$ under alternative hypothesis $H_{1}$, the GLRT decides $H_{1}$ if:(16)$$\begin{array}{c} L_{G}\left(x\right)=\frac{p(x;\hat{\theta_{1}},H_{1})}{p(x;\hat{\theta_{0}},H_{0})}>\gamma \end{array}$$
where $\hat{\theta_{1}}$ is the MLE of $\theta_{1}$ assuming $H_{1}$ is true, $\hat{\theta_{0}}$ is the MLE of $\theta_{0}$ assuming $H_{0}$ is true, and γ is the threshold.

### 6.2. Design of Test Statistic Based on Generalized Likelihood Ratio Test

Before going through the design process, it is beneficial to know the PDF of the measurement noise signal. This gives us enough knowledge to make the assumptions that are close to our realistic problem. Using the first part (*t* = 0–1 s) of the resultant residual in Figure 11, the PDF of the noise signal in a noise-only hypothesis is obtained and shown in Figure 12. The PDF of the noise signal in Figure 12 is very close to the PDF of a white Gaussian noise (WGN), thus it can be reasonably modeled with a WGN probability distribution function.

To design a realistic detector, it is assumed that the arrival time of the fault is completely unknown. Furthermore, the PDF is not completely known, meaning that the parameters mean μ and variance $\sigma^{2}$ are to be estimated using MLE. The noise in the resultant residual during operation in a healthy condition is modeled as WGN. Since the resultant residual ($R_{r}$) obtains a nonzero dc value when ITSC faults appear, the data are considered as only noise under nonfaulty hypothesis $H_{0}$, and an added dc level value to the noise under faulty hypothesis $H_{1}$. Thus, the detection problem becomes as follows:(17)$$\begin{array}{cc} H_{0}:x\left[n\right] & =w[n]\;n=0,1,...,N-1\\ H_{1}:x\left[n\right] & =A+w[n]\;n=0,1,...,N-1 \end{array}$$
where A is unknown amplitude with −∞<A<∞, and w[n] is WGN with unknown positive variance $0<\sigma^{2}<\infty$. The GLRT decides $H_{1}$ if:(18)$$\begin{array}{c} L_{G}\left(x\right)=\frac{p(x;\hat{A},{\hat{\sigma_{1}}}^{2},H_{1})}{p(x;{\hat{\sigma_{0}}}^{2},H_{0})}>\gamma \end{array}$$
where $\hat{A}$ and ${\hat{\sigma_{1}}}^{2}$ are the MLE of parameters *A* and $\sigma_{1}^{2}$ under $H_{1}$, and ${\hat{\sigma_{0}}}^{2}$ is the MLE of the parameter $\sigma_{0}^{2}$ under $H_{0}$. By maximizing $p(x;A,\sigma^{2},H_{1})$, parameters $\hat{A}$ and ${\hat{\sigma_{1}}}^{2}$ are obtained as follows [43]:(19)$$\begin{array}{cc} p(x;A,\sigma^{2},H_{1}) & =\frac{1}{{\left(2\pi \sigma^{2}\right)}^{\frac{N}{2}}}\exp\left[-\frac{1}{2\sigma^{2}}\sum_{N=0}^{N-1}{\left(x\left[n\right]-A\right)}^{2}\right]\\ \frac{\partial p(x;A,\sigma^{2},H_{1})}{\partial A} & =0\Rightarrow \hat{A}=\bar{x}\\ \frac{\partial p(x;A,\sigma^{2},H_{1})}{\partial \sigma_{1}^{2}} & =0\Rightarrow {\hat{\sigma_{1}}}^{2}=\frac{1}{N}\sum_{N=0}^{N-1}{\left(x\left[n\right]-A\right)}^{2} \end{array}$$
which results in:(20)$$\begin{array}{c} p\left(x;\hat{A},{\hat{\sigma_{1}}}^{2},H_{1}\right)=\frac{1}{{\left(2\pi {\hat{\sigma_{1}}}^{2}\right)}^{\frac{N}{2}}}\exp\left(-\frac{N}{2}\right) \end{array}$$

Similarly, by maximizing $p(x;{\hat{\sigma_{0}}}^{2},H_{0})$, ${\hat{\sigma_{0}}}^{2}$ is obtained as follows:(21)$$\begin{array}{cc} p(x;\sigma^{2},H_{0}) & =\frac{1}{{\left(2\pi \sigma^{2}\right)}^{\frac{N}{2}}}\exp\left(-\frac{1}{2\sigma^{2}}\sum_{N=0}^{N-1}x^{2}\left[n\right]\right)\\ \frac{\partial p(x;\sigma^{2},H_{0})}{\partial \sigma_{0}^{2}} & =0\Rightarrow {\hat{\sigma_{0}}}^{2}=\frac{1}{N}\sum_{N=0}^{N-1}x^{2}\left[n\right] \end{array}$$
which results in:(22)$$\begin{array}{c} p\left(x;{\hat{\sigma_{0}}}^{2},H_{0}\right)=\frac{1}{{\left(2\pi {\hat{\sigma_{0}}}^{2}\right)}^{\frac{N}{2}}}\exp\left(-\frac{N}{2}\right) \end{array}$$Therefore, (Equation 18) becomes:(23)$$\begin{array}{c} L_{G}\left(x\right)={\left(\frac{{\hat{\sigma_{0}}}^{2}}{{\hat{\sigma_{1}}}^{2}}\right)}^{\frac{N}{2}} \end{array}$$
which is equivalent to:(24)$$\begin{array}{c} 2lnL_{G}\left(x\right)=Nln\frac{{\hat{\sigma_{0}}}^{2}}{{\hat{\sigma_{1}}}^{2}} \end{array}$$From (Equation 19) and (Equation 21), ${\hat{\sigma_{1}}}^{2}$ can intuitively be obtained as follows:(25)$$\begin{array}{cc} {\hat{\sigma_{1}}}^{2} & =\frac{1}{N}\sum_{N=0}^{N-1}{\left(x\left[n\right]-A\right)}^{2}=\frac{1}{N}\sum_{N=0}^{N-1}{\left(x\left[n\right]-\bar{x}\right)}^{2}\\ \; & =\frac{1}{N}\sum_{N=0}^{N-1}\left(x{\left[n\right]}^{2}-2x\left[n\right]\bar{x}+{\bar{x}}^{2}\right)=\frac{1}{N}\sum_{N=0}^{N-1}x{\left[n\right]}^{2}-{\bar{x}}^{2}\\ \; & ={\hat{\sigma_{0}}}^{2}-{\bar{x}}^{2} \end{array}$$
which yields:(26)$$\begin{array}{c} 2lnL_{G}\left(x\right)=Nln\left(1+\frac{{\bar{x}}^{2}}{{\hat{\sigma_{1}}}^{2}}\right) \end{array}$$

Since $ln(1+\frac{{\bar{x}}^{2}}{{\hat{\sigma_{1}}}^{2}})$ is monotonically increasing with respect to $\frac{{\bar{x}}^{2}}{{\hat{\sigma_{1}}}^{2}}$, an equivalent and normalized test statistic can be obtained as follows:(27)$$\begin{array}{c} T\left(x\right)=\frac{{\bar{x}}^{2}}{{\hat{\sigma_{1}}}^{2}}>\gamma^{\prime } \end{array}$$

The GLRT has normalized the statistic by ${\hat{\sigma_{1}}}^{2}$ which allows the threshold to be determined. Since the PDF of T(x) under null hypothesis $H_{0}$ does not depend on $\sigma^{2}$, the threshold is independent of the value $\sigma^{2}$ [42].

### 6.3. GLRT for Large Data Records

As N⟶∞, the asymptotic PDFs of $\bar{x}$ will converge to normal distributions under both hypotheses as follows:(28)$$\begin{array}{c} \bar{x}∼\left\{\begin{array}{cc} N(0,\sigma^{2}) & \mathrm{under}\;H_{0}\\ N(A,\sigma^{2}) & \mathrm{under}\;H_{1} \end{array}\right. \end{array}$$
and therefore:(29)$$\begin{array}{c} \frac{\bar{x}}{\sigma }∼\left\{\begin{array}{cc} N(0,1) & \mathrm{under}\;H_{0}\\ N(\frac{A}{\sigma },1) & \mathrm{under}\;H_{1} \end{array}\right. \end{array}$$

Squaring the normalized statistic in (Equation 29) will lead to the modified test statistic T(x) in (Equation 27) which produces a central chi-squared distribution under $H_{0}$ and a noncentral chi-squared distribution under $H_{0}$, with one degree of freedom: (30)$$\begin{array}{c} T\left(x\right)=\frac{{\bar{x}}^{2}}{\sigma^{2}}∼\left\{\begin{array}{cc} X_{1}^{2} & \mathrm{under}\;H_{0}\\ X_{1}^{\prime }2\left(\lambda \right) & \mathrm{under}\;H_{1} \end{array}\right. \end{array}$$
where λ is the noncentrality parameter and is calculated as [42]:(31)$$\begin{array}{c} \lambda =\frac{A^{2}}{\sigma^{2}}=\frac{{\bar{x}}^{2}}{\sigma^{2}} \end{array}$$

It was shown in (Equation 30) that T(x) has a noncentral chi-squared distribution with one degree of freedom, and it is equal to the square of random variable *x* in (Equation 29), therefore $x∼N(\sqrt{\lambda },1)$. Thus, the probability of a false alarm ($P_{FA}$) can be obtained as:(32)$$\begin{array}{cc} P_{FA} & =Pr\{T\left(x\right)>\gamma^{\prime };H_{0}\}\\ \; & =Pr\left\{x>\sqrt{\gamma^{\prime }};H_{0}\right\}+Pr\left\{x<-\sqrt{\gamma^{\prime }};H_{0}\right\}\\ \; & =2Q(\sqrt{\gamma^{\prime }}) \end{array}$$
where Q(x) is the right-tail probability of random variable *x*. Thus, the threshold can be obtained as follows:(33)$$\begin{array}{c} \gamma^{\prime }={\left[Q^{-1}\left(\frac{P_{FA}}{2}\right)\right]}^{2} \end{array}$$

Similarly, the probability of detection $P_{D}$ can be obtained as follows:(34)$$\begin{array}{cc} P_{D} & =Pr\{T\left(x\right)>\gamma^{\prime };H_{1}\}\\ \; & =Pr\left\{x>\sqrt{\gamma^{\prime }};H_{1}\right\}+Pr\left\{x<-\sqrt{\gamma^{\prime }};H_{1}\right\}\\ \; & =Q\left(\sqrt{\gamma^{\prime }}-\sqrt{\lambda }\right)+Q\left(\sqrt{\gamma^{\prime }}+\sqrt{\lambda }\right)\\ \; & =Q\left(Q^{-1}\left(\frac{P_{FA}}{2}\right)-\sqrt{\lambda }\right)+Q\left(Q^{-1}\left(\frac{P_{FA}}{2}\right)+\sqrt{\lambda }\right) \end{array}$$

### 6.4. GLRT Test on Residual Response

For the case study, the statistical detector should be designed in a way that it is able to detect even the smallest ITSC fault (<1%). Therefore, the noncentrality parameter λ is calculated based on the implementation of (Equation 31) on the resultant residual at *t* = 4.471s–7.238 s when the motor is experiencing the lowest ITSC fault level in phase *a* winding and yields λ=6.78. Using this value, the threshold and receiver operating characteristics (ROC) of the detector is obtained based on (Equation 32)–(Equation 34) and shown in Figure 13. The $P_{FA}$ values here are for the lowest ITSC fault level in phase *a*, which means other ITSC faults in phases *b* and *c* have lower $P_{FA}$ values. Using $P_{FA}=2\%$, the threshold is obtained as $\gamma^{\prime }=5.41$, and this results in $P_{D}=60.93\%$ for ITSC in phase *a*. Furthermore, the probability of detection for ITSC faults in phases *b* and *c* and the encoder fault are calculated $P_{D}=98.13\%$, $P_{D}=100\%$, and $P_{D}=99.65\%$, respectively.

The test statistics were implemented on the resultant residual as shown in Figure 14. The values ${\bar{x}}^{2}$ and ${\hat{\sigma_{1}}}^{2}$ were calculated using a moving window (FIFO register) with the length of N=10,000, which runs through the resultant residual over time. Figure 14a shows the output of test statistic on resultant residual along with the threshold of $\gamma^{\prime }=5.41$ while Figure 14b shows the output of the test statistic on $R_{4}$. The test statistic’s output value is compared with the threshold value over time, and if it exceeds the threshold, the fault alarm is tripped accordingly. Figure 15 shows the detector’s logical output value which attains a low value in a healthy condition and a high value during a faulty case. This proves that the detector has successfully detected all the faults that are fairly close to expected values of $P_{D}$, while experiencing no false alarm.

## 7. Discussion

Some remarks can be withdrawn regarding the presented methodology and the obtained results. First, structural analysis for detecting ITSC and encoder faults was successfully implemented on the in-house setup including the PMSM and the drive system, and the residuals were formed based on ARRs. Second, a GLRT-based detector was designed to effectively detect the changes in the residuals even with unknown noise parameters. Third, a scientific threshold was calculated based on the probability of a false alarm ($P_{FA}$) and the probability of detection ($P_{D}$). The suggested combination method is very effective for the fault detection since it can detect the lowest level of ITSC fault, i.e., one single shorted turn (<1%) in the stator winding. On the other hand, using a Clarke transformation disabled the diagnostic system to isolate the ITSC faults in different phases, and using a moving window with the length of N=10,000 over the test statistics causes a delay in detection of the faults. These small demerits were found when testing the diagnostic method under the smallest ITSC fault.

In previous studies, a GLRT-based detector has been implemented for stator imbalancefault detection in induction motors [44]. The noise parameters were also considered unknown, and therefore, they have been replaced with their MLEs. Moreover, a threshold was calculated based on $P_{FA}=0.1\%$ and $P_{D}$, which makes the diagnostic system experience fewer false alarms. However, the first fault level that the system can detect is 25% of stator-phase resistance, which is a quite high level of fault severity. As a result, the system would go into severe imbalance from the time that the fault appears until the time the diagnostic system detects it. In our case, even if the $P_{FA}$ was chosen as 0.1%, the $P_{D}$ for ITSC in phase *a* would be 24.61%, the $P_{D}$ for ITSC in phase *b* would be 86.8%, the $P_{D}$ for ITSC in phase *c* would be 99.99%, and the $P_{D}$ for the encoder fault would be 95.86%. Thus, the diagnostic system still detects the smallest fault, even with $P_{FA}=0.1\%$. However, knowing that a slightly higher probability of a false alarm is not that irritating ($P_{FA}=2\%$), a better probability of detection is achieved ($P_{D}=60.93\%$) in our study based on setting a lower threshold. Other studies with different methods have also chosen a higher level of fault as the starting point. A Kalman filter for detection of ITSC in PM synchronous generators has been implemented in [45], which can successfully detect fault levels as low as 8%. In addition, a combination of an extended Park’s vector approach with spectral frequency analysis was introduced in [46] which could successfully detect three shorted turns in synchronous and induction motors.

## 8. Conclusions

This paper presents a novel method for real-time and effective detection of incipient ITSC and encoder faults in the PMSM. Structural analysis was employed to form the structural model of the PMSM. The Dulmage–Mendelsohn decomposition tool was used to evaluate the analytical redundancy of the structural model. The proposed diagnostic model was implemented on industrial PMSM, ITSC, and encoder faults were applied to the system in different time intervals, and residuals responses were obtained. Subsequently, a GLRT-based detector was designed and implemented based on the behavior of the residuals during healthy (only noise) and faulty (noise + signal) conditions. To make the GLRT-based detector effective to deal with such a realistic problem, the parameters such as mean μ and variance $\sigma^{2}$ in the probability density function of the noise signal were considered to be unknown. By replacing these unknown parameters by their maximum likelihood estimates, a test statistic was achieved for the GLRT-based ITSC and encoder fault detector. Following this step, a threshold was obtained based on choosing the probability of a false alarm $P_{FA}$ and the probability of detection $P_{D}$ for each detector based on which decision was made to indicate the presence of the fault. The experimental results show that the designed GLRT-based detector is able to efficiently detect even small ITSC and encoder faults in the presence of noise, proving the effectiveness of this diagnostic approach.

## Figures and Tables

**Figure 1 sensors-22-03407-f001:**
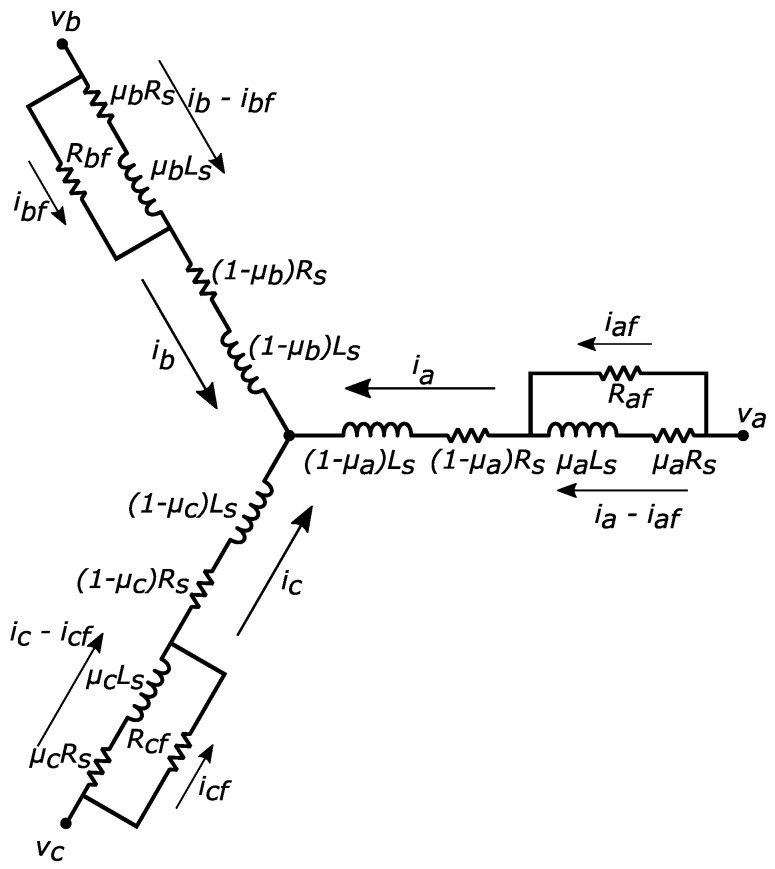
Schematic of PMSM stator windings under ITSC faults.

**Figure 2 sensors-22-03407-f002:**
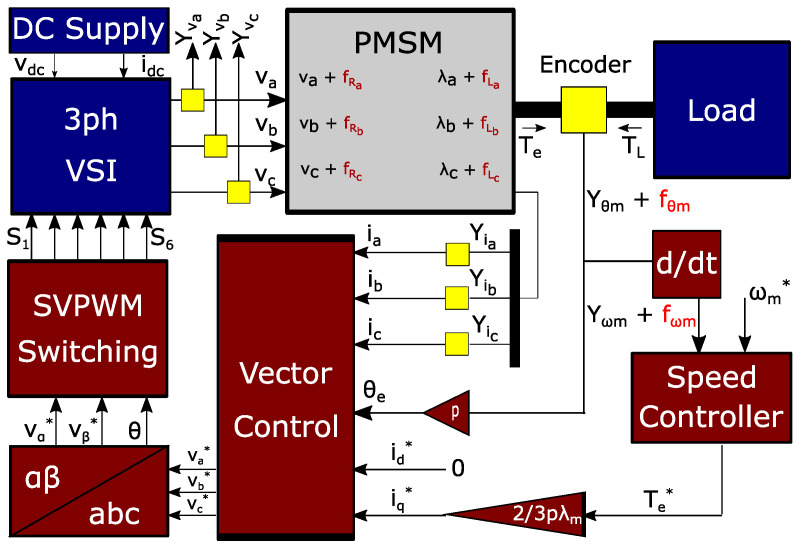
Modeling diagram of PMSM and drive system.

**Figure 3 sensors-22-03407-f003:**
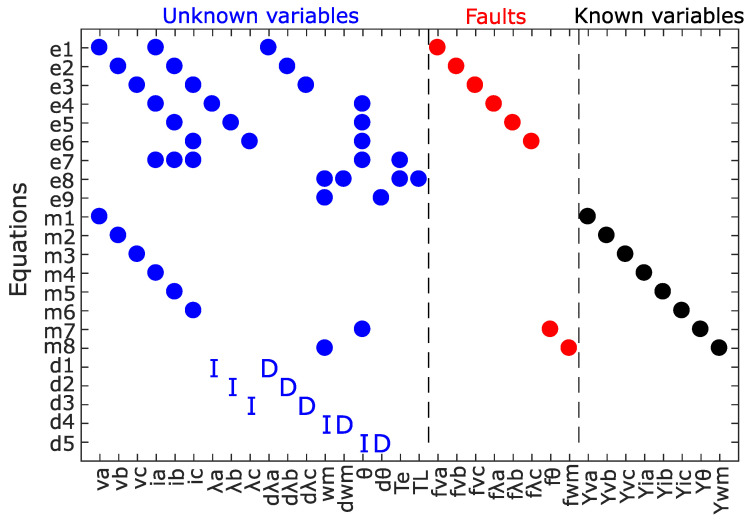
PMSM structural model.

**Figure 4 sensors-22-03407-f004:**
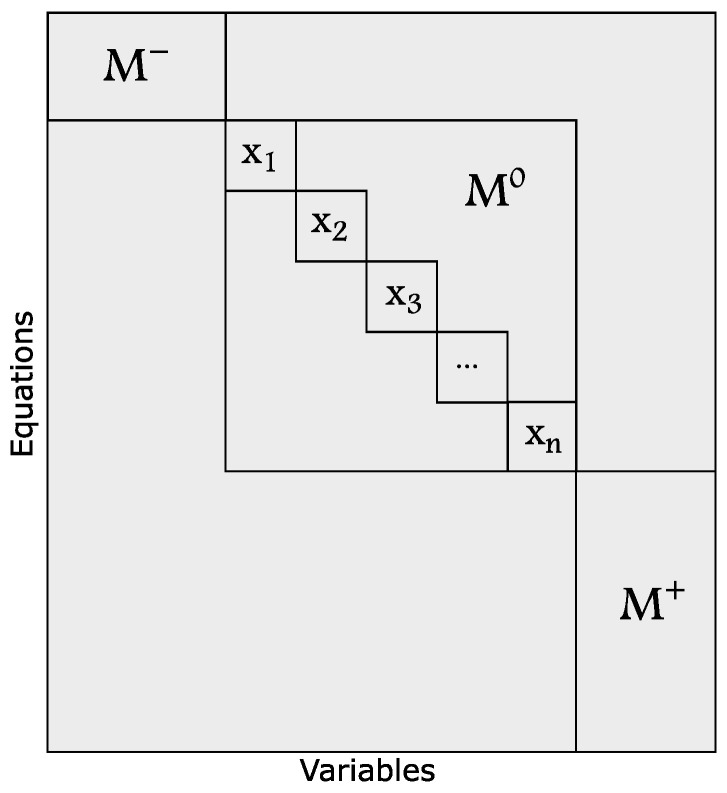
Canonical decomposition of the structure graph *M*.

**Figure 5 sensors-22-03407-f005:**
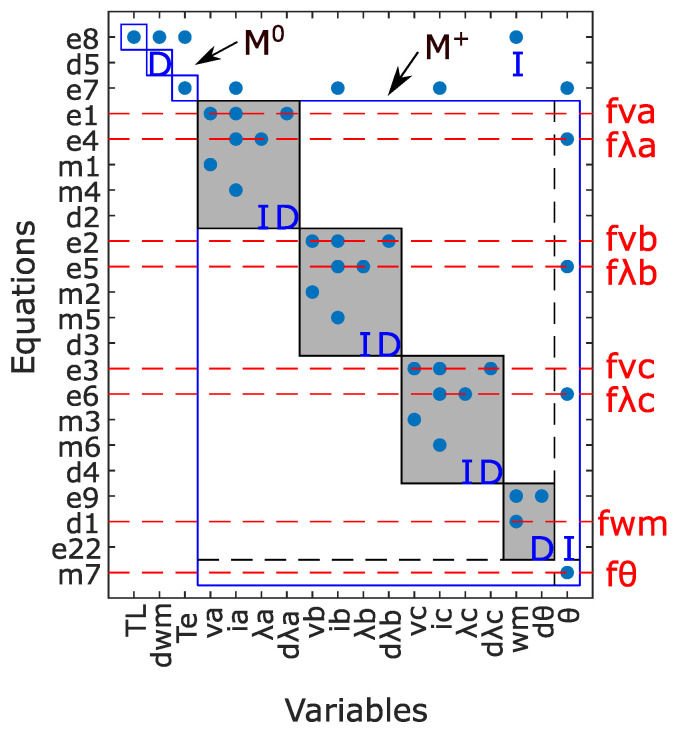
DM decomposition for PMSM structural model.

**Figure 6 sensors-22-03407-f006:**
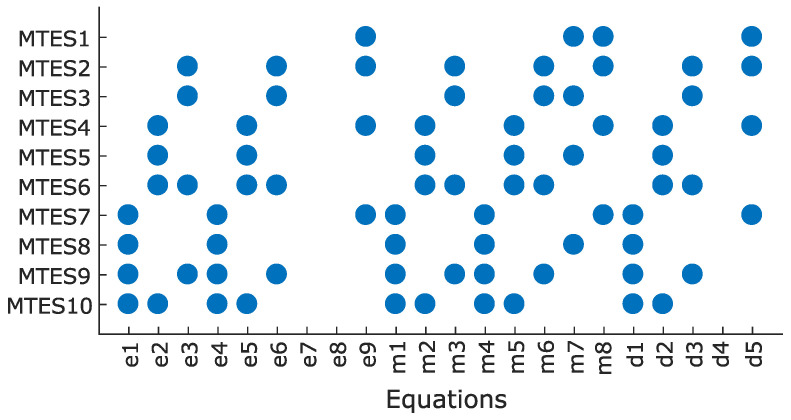
MTES sets.

**Figure 7 sensors-22-03407-f007:**
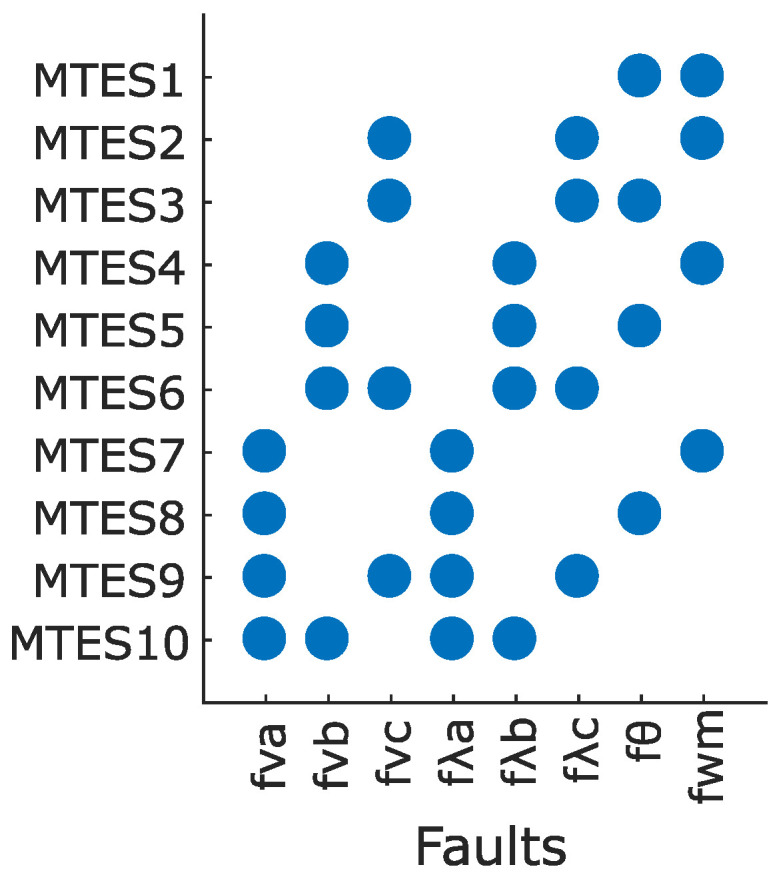
Fault signature matrix of MTES sets.

**Figure 8 sensors-22-03407-f008:**
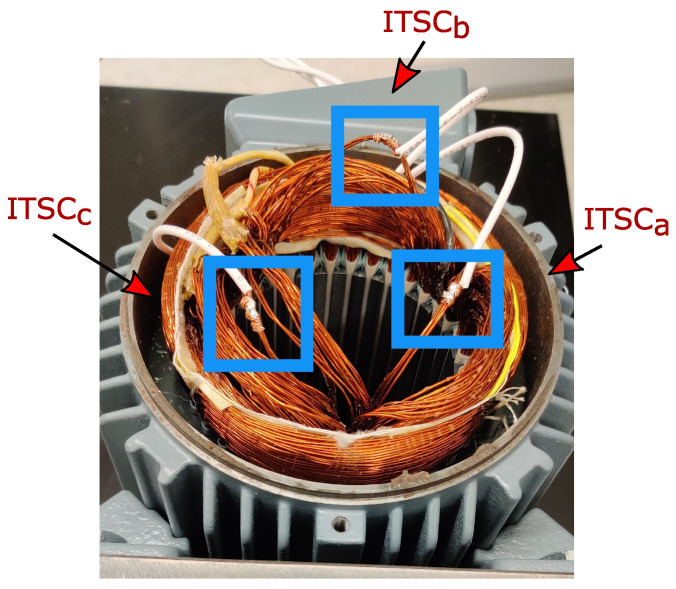
Applied ITSC faults on a PMSM.

**Figure 9 sensors-22-03407-f009:**
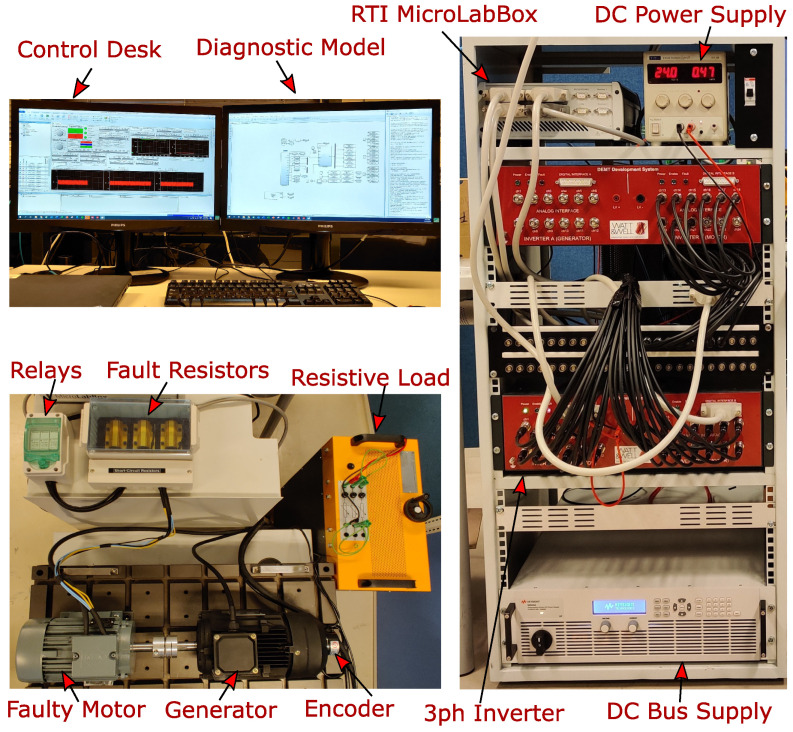
Experimental Setup for control and diagnosis of a PMSM.

**Figure 10 sensors-22-03407-f010:**
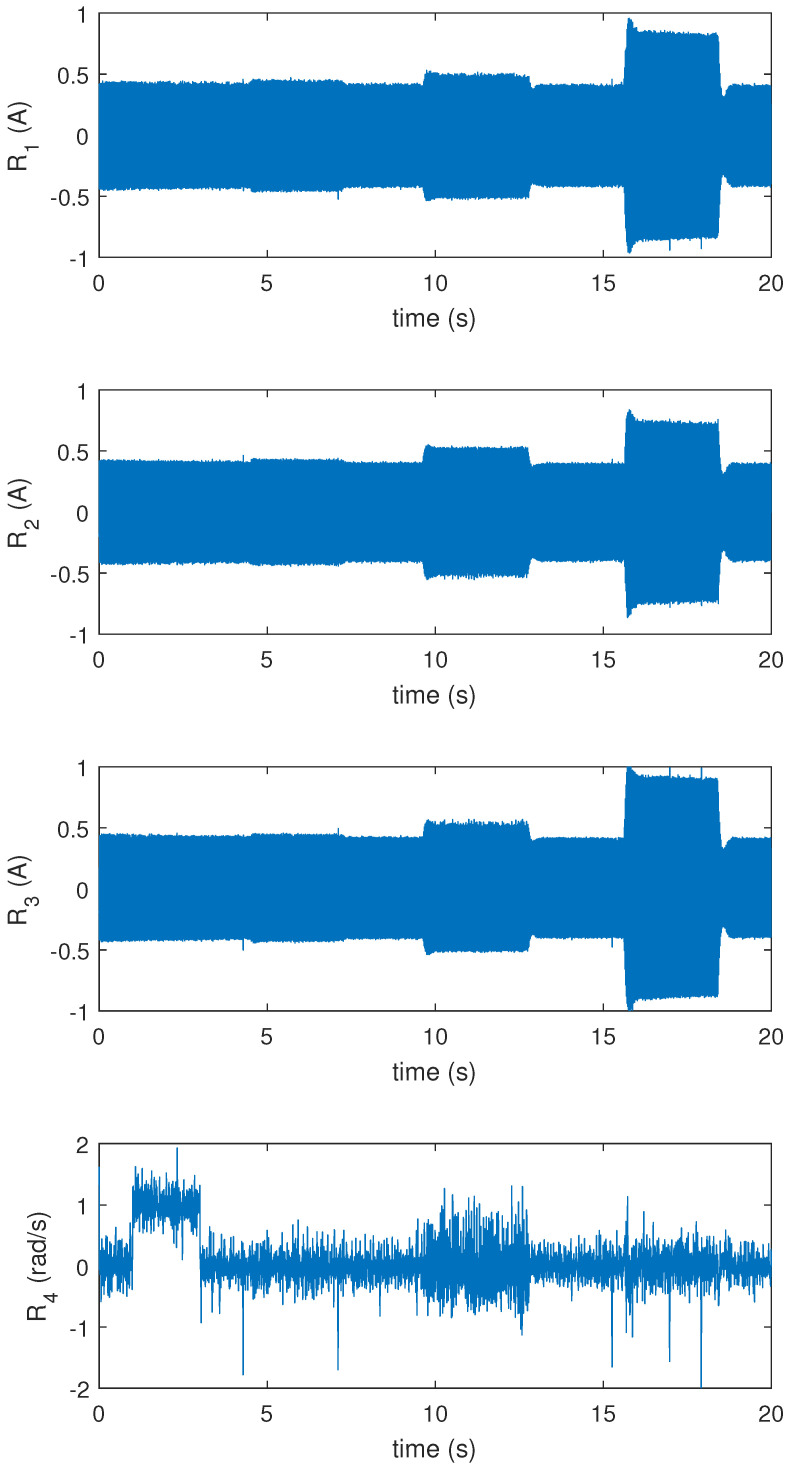
Residual responses in abc phases.

**Figure 11 sensors-22-03407-f011:**
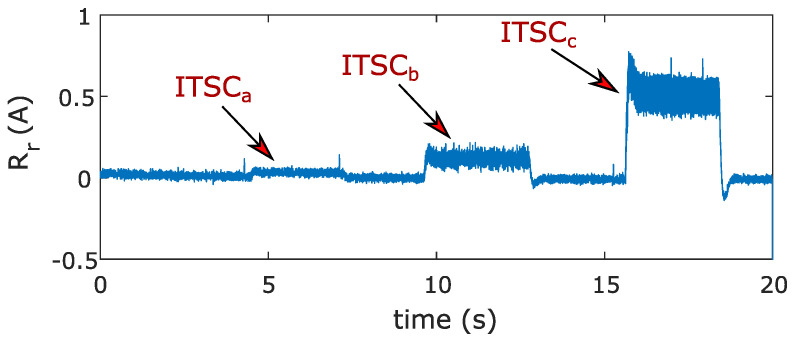
Resultant residual response in an αβ frame.

**Figure 12 sensors-22-03407-f012:**
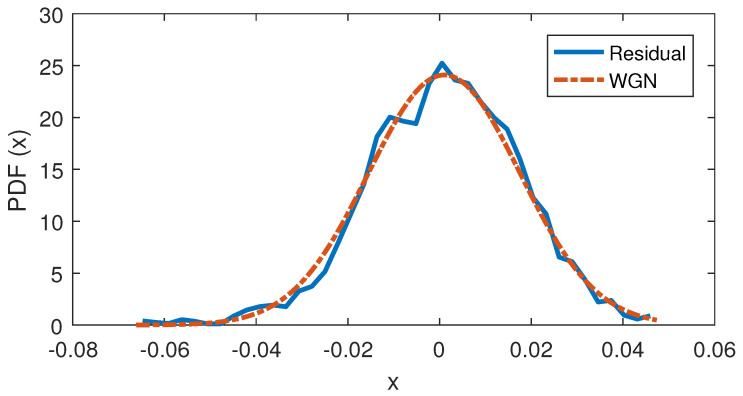
Comparison of PDF of residual and WGN.

**Figure 13 sensors-22-03407-f013:**
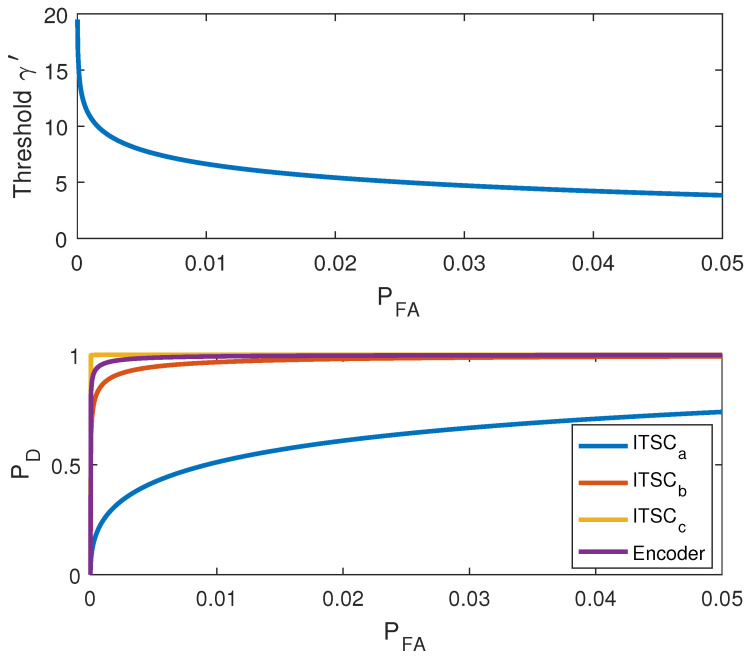
Threshold and ROC for low values of $P_{FA}$.

**Figure 14 sensors-22-03407-f014:**
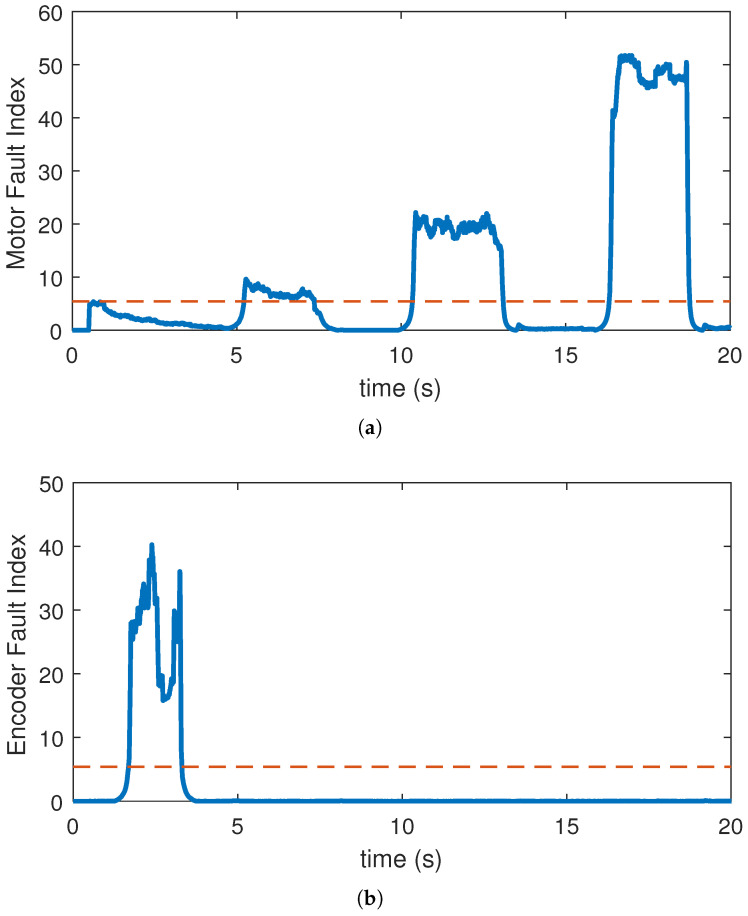
Test statistic for ITSC fault and encoder faults.

**Figure 15 sensors-22-03407-f015:**
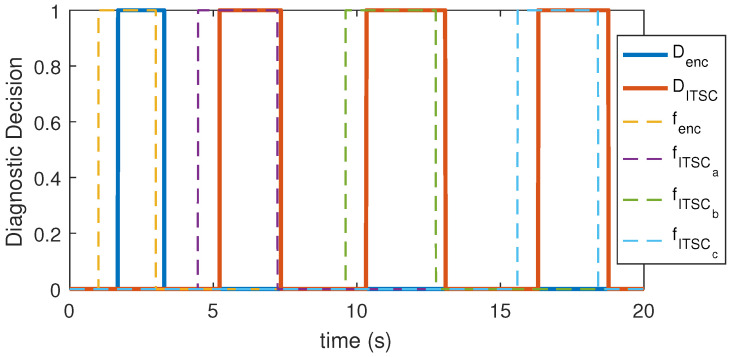
Timing of actual faults ($f_{enc},f_{ITSC_{a}},f_{ITSC_{b}},f_{ITSC_{c}}$) versus diagnostic system’s decision ($D_{enc},D_{ITSC}$).

**Table 1 sensors-22-03407-t001:** Parameters of a PM synchronous motor.

Symbol	Parameter	Value	Unit
$V_{dc}$	Rated dc bus voltage	280	V
$I_{s}$	Rated rms phase current	5	A
$T_{out}$	Rated Output Torque	7	N·m
$n_{s}$	Rated speed	1500	rpm
$R_{s}$	Phase resistance	0.8	Ω
$L_{s}$	Stator inductance	8.5	mH
*J*	Rotor inertia	0.0026	kg · m^2^
*b*	Rotor damping factor	0.00382	N · m · s/rad
$\lambda_{m}$	Flux linkage of PMs	0.3509	Wb-turn
*p*	Pole-pairs	2	

## Data Availability

The data presented in this study are available on request from the corresponding author. The data are not publicly available due to ongoing research within Ph.D. program.

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
