# Peer review of "Real-Time Detection of Incipient Inter-Turn Short Circuit and Sensor Faults in Permanent Magnet Synchronous Motor Drives Based on Generalized Likelihood Ratio Test and Structural Analysis"

_sensors, 2022, doi:10.3390/s22093407_

Round 1

Reviewer 1 Report

This paper investigates the fault detection of the permanent magnet synchronous motor drives. Both structural analysis and generalized likelihood ratio test are used. The work is meaningful. Some suggestions might be useful:

1. In Introduction, the analyses and challenges about the general model-based FD literatures might not be much adequate to support the authors' choice for the structural analysis.

2. In equation (1), the faults were added in e1 to e6, how about e7 to e9 where the torque, angular acceleration, and angular velocity might also be affected by the faults? Are there any faults that could be encountered by the measured voltages and currents? 

3. Based on the contributions given in lines 61-68 of page 2, not any innovation related to the structural analysis and GLRT was presented. The innovations about the methods may help the readers to understand deeply about the authors' work.

4. In subsection 4.1, how to know that there is only one redundancy of an MTES?

5. In subsection 4.2, the way to choose MTESs was not much clear.

6. The close combination of the structural analysis and GLRT was not very clear in section 6 in the reviewer's view.

7. Figure 11 only gave one second PDF of the resultant residual, was it enough for supporting the conclusion of WGN?

Reviewer 2 Report

This paper presents a very interesting study on fault detection in PMSM drive. However, there are several concerns that needs to be addressed:

  1. The authors claim that the signal-based methods face challenges of real-time implementations due to computational burden. Can you please justify the proposed method is better than the signal-based methods with regard to computational burden?
  2. Can you please elaborate on 3.3? It is quite difficult to relate DM decomposition to the detection of faults. I could not understand how  the DM decomposition model in figure 4 is obtained from the structural model in figure 3. More background information as well as detailed analysis should be provided.
  3. No clear explanation for MTES is given. I could hardly understand what MTES is about. Please also show clear definition for dianosability index.

Reviewer 3 Report

The manuscript presents an interesting research on ITSC fault detection in PMSM. Inter-Turn Short Circuit is a topic correctly addressed. Authors claim to obtain a high sensible methodology that allows detection with less than 1% of the winding short circuited.  

The manuscript starts with an accurate state of the art. Providing analysis on bibliography identifying applications and techniques.

The mathematical analysis is adequately presented, it introduces fault terms fva, fvb, fvc, fλa... the use of these fault terms is interesting for the study since it aims to obtain Analytic Redundant Relations. Combining Dulmage-Mendelsohn (DM) decomposition and Minimal Test Equation Support (MTES) authors relates fault terms with specific MTES sets. 

The authors introduces the test bench analysis on section 5. The test bench is sufficiently detailed. The experimental results shows the evolution of Residuals on a time series basis applying 1% - 3% and 5% ITSC faults in different periods of time. Residual response is adequately analyzed. 

Finally GRLT Test is used to analyze Residual REsponse.

The manuscript is clear and perfectly addressed. It is in my opinion ready for publication.

Author Response

Dear reviewer,

Thank you very much for taking the time to read and investigate the draft so thoroughly. We appreciate your positive feedback and wish you all the best.

Kind regards,
Corresponding author

Round 2

Reviewer 2 Report

Thanks for your effort in improving your paper. I would recommend for publication.